# Differentially Private Covariance Revisited

**Wei Dong, Yuting Liang, Ke Yi**
{wdongac,yliangbs,yike}@cse.ust.hk
Department of Computer Science
Hong Kong University of Science and Technology

## Abstract

In this paper, we present two new algorithms for covariance estimation under concentrated differential privacy (zCDP). The first algorithm achieves a Frobenius error of $\tilde{O}(d^{1/4}\sqrt{\mathrm{tr}}/\sqrt{n} + \sqrt{d}/n)$, where $\mathrm{tr}$ is the trace of the covariance matrix. By taking $\mathrm{tr} = 1$, this also implies a worst-case error bound of $\tilde{O}(d^{1/4}/\sqrt{n})$, which improves the standard Gaussian mechanism's $\tilde{O}(d/n)$ for the regime $d > \tilde{\Omega}(n^{2/3})$. Our second algorithm offers a tail-sensitive bound that could be much better on skewed data. The corresponding algorithms are also simple and efficient. Experimental results show that they offer significant improvements over prior work.

## 1 Introduction

Consider a dataset represented by a matrix $\mathbf{X} \in \mathbb{R}^{d \times n}$, where each column $X_i, i = 1, \ldots, n$ corresponds to an individual's information. As standard in the literature, we assume that all the $X_i$'s live in $\mathcal{B}_d$, the $d$-dimensional $\ell_2$-unit ball centered at the origin. In this paper, we revisit the problem of estimating the (empirical) covariance matrix $\mathbf{\Sigma}(\mathbf{X}) := \frac{1}{n}\sum_i X_i X_i^T = \frac{1}{n}\mathbf{X}\mathbf{X}^T$ under differential privacy (DP), a fundamental problem in high-dimensional data analytics and machine learning that requires little motivation. We often write $\mathbf{\Sigma}(\mathbf{X})$ as $\mathbf{\Sigma}$ when the context is clear. As with most prior work, we use the Frobenius norm $\|\tilde{\mathbf{\Sigma}} - \mathbf{\Sigma}\|_F$ to measure the error of the estimated covariance $\tilde{\mathbf{\Sigma}}$. To better focus, in the introduction we state all results under *concentrated different privacy (zCDP)* [10]; extensions of our results to pure-DP are given in Appendix I.

### 1.1 A Trace-sensitive Algorithm

For any symmetric matrix $\mathbf{A}$, we use $\mathbf{P}[\mathbf{A}]$ and $\mathbf{\Lambda}[\mathbf{A}]$ to denote its matrices of eigenvectors and eigenvalues, respectively, such that $\mathbf{A} = \mathbf{P}[\mathbf{A}]\mathbf{\Lambda}[\mathbf{A}]\mathbf{P}[\mathbf{A}]^T$; we use $\lambda_i[\mathbf{A}]$ to denote its $i$th largest eigenvalue. When $\mathbf{A} = \mathbf{\Sigma} = \mathbf{\Sigma}(\mathbf{X})$, we simply write $\mathbf{P} = \mathbf{P}[\mathbf{\Sigma}], \mathbf{\Lambda} = \mathbf{\Lambda}[\mathbf{\Sigma}], \lambda_i = \lambda_i[\mathbf{\Sigma}]$, so that $\mathbf{\Lambda} = \mathrm{diag}(\lambda_1, \cdots, \lambda_d)$ and $\mathbf{\Sigma} = \mathbf{P}\mathbf{\Lambda}\mathbf{P}^T$. Let $\mathbf{P} = [P_1 \; P_2 \; \cdots \; P_d]$, where $P_i$ is the orthonormal basis vector corresponding to $\lambda_i$. Rudimentary linear algebra yields $\lambda_k = \frac{1}{n}\sum_i (P_k^T X_i)^2$ for $1 \leqslant k \leqslant d$ and $||X_i||_2^2 = \sum_k (P_k^T X_i)^2$ for $1 \leqslant i \leqslant n$. Thus, it follows that

$$\mathrm{tr}[\mathbf{\Sigma}] = \mathrm{tr}[\mathbf{\Lambda}] = \sum_k \lambda_k = \sum_k \frac{1}{n}\sum_i (P_k^T X_i)^2 = \frac{1}{n}\sum_i\sum_k (P_k^T X_i)^2 = \frac{1}{n}\sum_i ||X_i||_2^2.$$

That is, $0 \leqslant \mathrm{tr}[\mathbf{\Lambda}] \leqslant 1$ is the average $\ell_2$ norm (squared) of the $X_i$'s, and we simply write it as $\mathrm{tr}$.

Recall that it is assumed that all the $X_i$'s live in $\mathcal{B}_d$. In practice, this is enforced by assuming an upper bound $B$ on the norms and scaling down all $X_i$ by $B$. As one often uses a conservatively large $B$, typical values of $\mathrm{tr}$ can be much smaller than 1, so a trace-sensitive algorithm would be

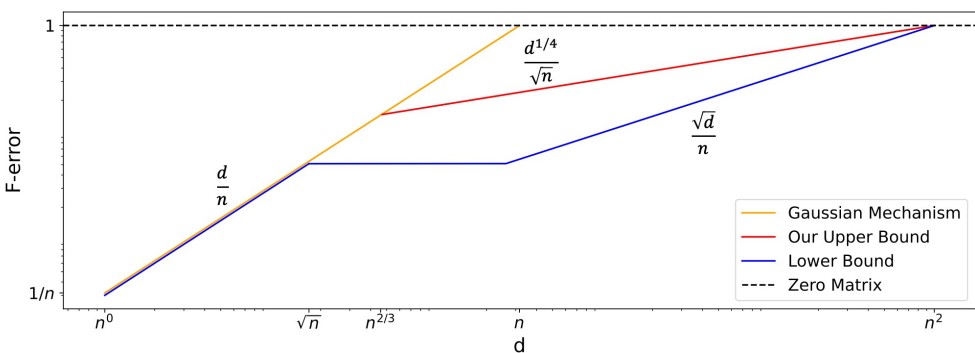

Figure 1: Currently known worst-case error bounds (both axes are in log scale).

more desirable. Indeed, Amin et al. [2] take this approach, describing an algorithm with error[1] $\tilde{O}(d^{3/4}\sqrt{\mathrm{tr}}/\sqrt{n} + \sqrt{d}/n)$ under zCDP[2]. Note that the $\sqrt{d}/n$ term inherits from mean estimation and the first term is the "extra" difficulty for covariance estimation. In this paper, we improve this term to $d^{1/4}\sqrt{\mathrm{tr}}/\sqrt{n}$ (we have a similar, albeit lesser, improvement under pure-DP; see Appendix I). Our algorithm is very simple: We first estimate $\mathbf{\Lambda}$ using the Gaussian mechanism (this is the same as in [2, 22]), then we estimate $\mathbf{P}$ by doing an eigendecomposition of $\mathbf{\Sigma}$ masked with Gaussian noise. Intuitively, we obtain a $\sqrt{d}$-factor improvement over the iterative methods of [2, 22], because we can obtain all eigenvectors from one noisy $\mathbf{\Sigma}$, while the iterative methods must allocate the privacy budget to all $d$ eigenvectors. Our algorithm is also more efficient, performing just two eigendecompositions and one matrix multiplication, whereas the algorithm in [2, 22] needs $O(d)$ such operations.

**Implication to worst-case bounds.** Covariance matrix has also been studied in the traditional worst-case setting, i.e., the bound should only depend on $d$ and $n$. Dwork et al. [17] show that the $\ell_2$-sensitivity of $\mathbf{\Sigma}$, i.e., $\max_{\mathbf{X} \sim \mathbf{X}'} \|\mathbf{\Sigma}(\mathbf{X}) - \mathbf{\Sigma}(\mathbf{X}')\|_F$ where $\mathbf{X} \sim \mathbf{X}'$ denotes two neighboring datasets that differ by one column, is $O(1/n)$. Thus, the standard *Gaussian mechanism* achieves an error of $\tilde{O}(d/n)$ by adding an independent Gaussian noise of scale $\tilde{O}(1/n)$ to each of the $d^2$ entries of $\mathbf{\Sigma}$. By taking $\mathrm{tr} = 1$, our trace-sensitive bound degenerates into $\tilde{O}(d^{1/4}/\sqrt{n})$. Note that the $\sqrt{d}/n$ term is dominated by $d^{1/4}/\sqrt{n}$ for $d < \tilde{O}(n^2)$, which is the parameter regime that allows non-trivial utility (i.e., the error is less than 1).

To better understand the situation, it is instructive to compare covariance estimation with mean estimation (where data are also drawn from the $\ell_2$ unit ball and the error is measured in $\ell_2$ norm), as the hardness of covariance estimation lies between $d$-dimensional mean estimation (only estimating the diagonal entries of $\mathbf{\Sigma}$) and $d^2$-dimensional mean estimation (treating $\mathbf{\Sigma}$ as a $d^2$-dimensional vector). This observation implies a lower bound $\tilde{\Omega}(\sqrt{d}/n)$ following from the same lower bound for mean estimation [19][3], and an upper bound $\tilde{O}(d/n)$ attained by the Gaussian mechanism. For $d < O(\sqrt{n})$, Kasiviswanathan et al. [23] prove a higher lower bound[4] $\tilde{\Omega}(d/n)$, which means that the complexity of covariance estimation is same as $d^2$-dimensional mean estimation in the low-dimensional regime, so one cannot hope to beat the Gaussian mechanism for small $d$. However, in the high-dimensional regime, our result indicates that the covariance problem is strictly easier, due to the correlations of the $d^2$ entries of $\mathbf{\Sigma}$. Another interesting consequence is that our error bound has utility for $d$ up to $\tilde{O}(n^2)$ (utility is lost when the error is $\tilde{O}(1)$, as returning a zero matrix can already achieve this error). This is the highest $d$ that allows for any utility, since even mean estimation requires $d < \tilde{O}(n^2)$ to have utility under zCDP [19, 10]. We pictorially show the currently known

---

[1]We use the $\tilde{O}$ notation to suppress the dependency on the privacy parameters and all polylogarithmic factors. We use $e$ as the base of log (unless stated otherwise) and define $\log(x) = 1$ for any $x \leqslant e$.

[2]Their paper states the error bound under pure-DP and for estimating $\mathbf{X}\mathbf{X}^T$ (i.e., without normalization by $1/n$); we show how this bound is derived from their result in Appendix C.

[3]This paper proves the lower bound under the statistical setting; in Appendix D, we show how it implies the claimed lower bound under the empirical setting.

[4]Their lower bound is under approximate-DP, which also holds under zCDP.

(worst-case) upper and lower bounds in Figure 1. It remains an interesting open problem to close the gap for $\tilde{\Omega}(\sqrt{n}) < d < \tilde{O}(n^2)$.

Through private communication with Aleksandar Nikolov, it is observed that the *projection mechanism* [31, 15] can also be shown to have error $\tilde{O}(d^{1/4}/\sqrt{n})$ when applied to the covariance problem. In Appendix E, we make this connection more explicit, while also giving an efficient implementation. However, the projection mechanism is not trace-sensitive.

## 1.2 A Tail-sensitive Algorithm

A trace-sensitive bound only makes use of the average $\ell_2$ norm, which cannot capture the full distribution. Next, we design an algorithm with an error bound that more closely depends on the distribution of the norms. We characterize this distribution using the $\tau$-*tail* ($\mathbb{I}(\cdot)$ is the indicator function):

$$\gamma(\mathbf{X}, \tau) = \frac{1}{n} \sum_i \|X_i\|_2^2 \cdot \mathbb{I}(\|X_i\|_2 > \tau), \tau \in [0, 1]. \tag{1}$$

Note that $\gamma(\mathbf{X}, \tau)$ decreases as $\tau$ increases. In particular, $\gamma(\mathbf{X}, 0) = \mathrm{tr}, \gamma(\mathbf{X}, 1) = 0$.

A common technique to reduce noise, at the expense of some bias, is to clip all the $X_i$'s so that they have norms at most $\tau$, for some threshold $\tau$. This yields an error of $\mathrm{Noise}(\mathbf{X}, \tau) + \gamma(\mathbf{X}, \tau)$, where $\mathrm{Noise}(\mathbf{X}, \tau)$ denotes the error bound of the mechanism when all the $X_i$'s have norm bounded by $\tau$, and $\gamma(\mathbf{X}, \tau)$ is the (additional) bias caused by clipping. Opting for the better of the Gaussian mechanism or our trace-sensitive mechanism, we have

$$\mathrm{Noise}(\mathbf{X}, \tau) = \tilde{O}\left( \min\left( \frac{\tau^2 d}{n}, \frac{\tau d^{1/4}\sqrt{\mathrm{tr}}}{\sqrt{n}} + \frac{\tau^2\sqrt{d}}{n} \right) \right). \tag{2}$$

The technical challenge is therefore choosing a good $\tau$ in a differentially private manner. We design a DP mechanism to choose the optimal $\tau$ up to a polylogarithmic multiplicative factor and an exponentially small additive term. It also adaptively selects the better of Gaussian mechanism or the trace-sensitive mechanism depending on the relationship between $d, n$, and a privatized $\mathrm{tr}$. More precisely, our adaptive mechanism achieves an error of

$$\tilde{O}\left( \min_{\tau} \left( \mathrm{Noise}(\mathbf{X}, \tau) + \gamma(\mathbf{X}, \tau) \right) + 2^{-dn} \right). \tag{3}$$

Note that this tail-sensitive bound is always no worse (modulo the $2^{-dn}$ term) than $\mathrm{Noise}(\mathbf{X}, 1)$ (i.e., without clipping), and can be much better for certain norm distributions. In particular, the tail-sensitive bound would work very well on many real datasets with skewed distributions, e.g., most data vectors have small norms with a few having large norms. For example, suppose $d = n^{3/4}$, and a constant number of data vectors have $\ell_2$ norm 1 while the others have norm $n^{-1/4}$. Then $\sqrt{\mathrm{tr}} = \Theta(n^{-1/4})$, so $\mathrm{Noise}(\mathbf{X}, 1)$ takes the trace-sensitive bound, which is $\tilde{O}(n^{-9/16})$. On the other hand, (3) is at most $\tilde{O}(n^{-13/16})$ by taking $\tau = n^{-1/4}$.

## 2 Related Work

Mean estimation and covariance estimation are perhaps the most fundamental problems in statistics and machine learning, and how to obtain the best estimates while respecting individual's privacy has attracted a lot of attention in recent years. Mean estimation under differential privacy is now relatively well understood, with the optimal worst-case error being $\tilde{\Theta}(\sqrt{d}/n)$ [19], achieved by the standard Gaussian mechanism [17]. In contrast, the covariance problem is more elusive. As indicated in Figure 1, its complexity is probably a piecewise linear (in the log-log scale) function.

When most data have norms much smaller than the upper bound given *a priori*, the worst-case bounds above are no longer optimal. In these cases, it is more desirable to have an error bound that is instance-specific. Clipping is a common technique for mean estimation [3, 18, 4, 34, 29] and it is known that running the Gaussian mechanism after clipping $\mathbf{X}$ with a certain quantile of the norms of the $X_i$'s achieves instance-optimality in a certain sense [3, 18]. However, for covariance estimation, we show in Appendix F that no quantile can be the optimal clipping threshold achieving the bound in

(3). Nevertheless, the bound in (3) is only achieving the optimal clipping threshold; we cannot say that is instance-optimal, since $\text{Noise}(\cdot)$ is not even known to be worst-case optimal.

Closely related to covariance estimation are the PCA problem and low-rank approximation. Instead of finding all eigenvalues and eigenvectors, they only aim at finding the largest one or a few. For these problems, iterative methods [2, 22, 38, 17, 11, 36] should perform better than the Gaussian mechanism or our algorithm, both of which try to recover the full covariance matrix.

Many covariance estimation algorithms have been proposed under the statistical setting, where the $X_i$'s are i.i.d. samples drawn from a certain distribution, e.g., a multivariate Gaussian [19, 9, 8, 1, 21, 28, 5, 25]. Instead of the Frobenius error, many of them adopt the Mahalanobis error $\|\widetilde{\boldsymbol{\Sigma}} - \boldsymbol{\Sigma}\|_{\boldsymbol{\Sigma}} := \|\boldsymbol{\Sigma}^{-1/2}\widetilde{\boldsymbol{\Sigma}}\boldsymbol{\Sigma}^{-1/2} - \mathbf{I}\|_F$, which can be considered as a normalized version of the former. It is known that $\lambda_d\|\mathbf{A} - \boldsymbol{\Sigma}\|_{\boldsymbol{\Sigma}} \leqslant \|\mathbf{A} - \boldsymbol{\Sigma}\|_F \leqslant \lambda_1\|\mathbf{A} - \boldsymbol{\Sigma}\|_{\boldsymbol{\Sigma}}$, so when $\boldsymbol{\Sigma}_{\mathbb{D}}$ is well-conditioned, i.e., $\lambda_1/\lambda_d = O(1)$, any Frobenius error directly translates to a Mahalanobis error. However, for the Mahalanobis error, the more challenging question is how to deal with an ill-conditioned $\boldsymbol{\Sigma}$, for which [19, 8] have provided elegant solutions for the case where $\mathbb{D}$ is a multivariate Gaussian. It would be interesting to see if their methods can be combined with the tail-sensitive techniques in this paper to solve this problem for other distribution families, in particular, heavy-tailed distributions. For the lower bound, very recently, Kamath et al. [20] proved a similar lower bound for the low-dimensional regime as in [23] but under the statistical setting.

## 3 Preliminaries

### 3.1 Differential Privacy

We say that $\mathbf{X}, \mathbf{X}' \in \mathbb{R}^{d \times n}$ are neighbors if they differ by one column, denoted $\mathbf{X} \sim \mathbf{X}'$.

**Definition 1** (Differential Privacy (DP) [16]). *For $\varepsilon > 0$ and $\delta \geqslant 0$, a randomized mechanism $\mathcal{M} : \mathbb{R}^{d \times n} \to \mathcal{Y}$ satisfies $(\varepsilon, \delta)$-DP if for any $\mathbf{X} \sim \mathbf{X}'$ and any $\mathcal{S} \subseteq \mathcal{Y}$, $\Pr[\mathcal{M}(\mathbf{X}) \in \mathcal{S}] \leqslant e^\varepsilon \cdot \Pr[\mathcal{M}(\mathbf{X}') \in \mathcal{S}] + \delta$.*

In particular, we call it *pure-DP* if $\delta = 0$; otherwise *approximate-DP*.

**Definition 2** (Concentrated Differential Privacy (zCDP) [10]). *For $\rho > 0$, a randomized mechanism $M : \mathbb{R}^{d \times n} \to \mathcal{Y}$ satisfies $\rho$-zCDP if for any $\mathbf{X} \sim \mathbf{X}'$, $D_\alpha\left(\mathcal{M}(\mathbf{X})\|\mathcal{M}(\mathbf{X}')\right) \leqslant \rho \cdot \alpha$ for all $\alpha > 1$, where $D_\alpha\left(\mathcal{M}(\mathbf{X})\|\mathcal{M}(\mathbf{X}')\right)$ is the $\alpha$-Rényi divergence between $\mathcal{M}(\mathbf{X})$ and $\mathcal{M}(\mathbf{X}')$.*

The relationship between these DP definitions is as follows. Pure-DP, also written as $\varepsilon$-DP, implies $\frac{\varepsilon^2}{2}$-zCDP, which further implies $\left(\frac{\varepsilon^2}{2} + \varepsilon\sqrt{2\log\frac{1}{\delta}}, \delta\right)$-DP for any $\delta > 0$.

To preserve $\varepsilon$-DP for a query $Q$, a standard mechanism is to add independent Laplace noises with scale proportional to the (global) $\ell_1$-sensitivity of $Q$ to each dimension.

**Lemma 1** (Laplace Mechanism [13]). *Given $Q : \mathbb{R}^{d \times n} \to \mathbb{R}^k$, let $\text{GS}_Q := \max_{\mathbf{X} \sim \mathbf{X}'} \|Q(\mathbf{X}) - Q(\mathbf{X}')\|_1$. The mechanism $\mathcal{M}(\mathbf{X}) = Q(\mathbf{X}) + \frac{\text{GS}_Q}{\varepsilon} \cdot \mathbf{Y}$ where $\mathbf{Y} \sim \text{Lap}(1)^k$, preserves $\varepsilon$-DP.*

The following composition property of $\varepsilon$-DP allows us to design algorithms in a modular fashion.

**Lemma 2** (Basic Composition). *If $\mathcal{M}$ is an adaptive composition of mechanisms $\mathcal{M}_1, \mathcal{M}_2, \ldots, \mathcal{M}_t$, where each $\mathcal{M}_i$ satisfies $\varepsilon_i$-DP, then $\mathcal{M}$ satisfies $(\sum_i \varepsilon_i)$-DP.*

For $\rho$-zCDP, the standard method is the *Gaussian mechanism*:

**Lemma 3** (Gaussian Mechanism [10]). *Given $Q : \mathbb{R}^{d \times n} \to \mathbb{R}^k$, let $\text{GS}_Q := \max_{\mathbf{X} \sim \mathbf{X}'} \|Q(\mathbf{X}) - Q(\mathbf{X}')\|_2$. The mechanism $\mathcal{M}(\mathbf{X}) = Q(\mathbf{X}) + \frac{\text{GS}_Q}{\sqrt{2\rho}} \cdot \mathbf{Y}$ where $\mathbf{Y} \sim \mathcal{N}\left(0, \mathbf{I}_{k \times k}\right)$, preserves $\rho$-zCDP.*

It has been shown that the covariance matrix has an $\ell_2$-sensitivity of $\frac{\sqrt{2}}{n}$ [8]. Thus, the Gaussian mechanism for covariance, denoted GaussCov, simply adds an independent Gaussian noise with scale $\frac{1}{\sqrt{\rho}n}$ to each entry of $\boldsymbol{\Sigma}$. Considering that $\boldsymbol{\Sigma}$ is symmetric, symmetric noises also suffice, which preserve the symmetry of the privatized $\boldsymbol{\Sigma}$. More precisely, we draw a random noise matrix $\mathbf{W}$ where $w_{j,k} \sim \mathcal{N}(0, 1)$ i.i.d. for $1 \leqslant j \leqslant k \leqslant d$ and $w_{k,j} = w_{j,k}$, denoted as $\mathbf{W} \sim \text{SGW}(d)$. Then GaussCov outputs $\widetilde{\boldsymbol{\Sigma}}_{\text{Gau}} = \boldsymbol{\Sigma} + \frac{1}{\sqrt{\rho}n} \cdot \mathbf{W}$.

A similar composition property exists for $\rho$-zCDP.

**Lemma 4** (Composition Theorem [10]). *If $\mathcal{M}$ is an adaptive composition of algorithms $\mathcal{M}_1, \mathcal{M}_2, \ldots, \mathcal{M}_t$, where each $\mathcal{M}_i$ satisfies $\rho_i$-zCDP, then $\mathcal{M}$ satisfies $(\sum_i \rho_i)$-zCDP.*

### 3.2 The Sparse Vector Technique

The *Sparse Vector Technique* (SVT) [14] has as input a sequence of scalar queries, $f_1(\mathbf{X}), f_2(\mathbf{X}), \ldots, f_t(\mathbf{X})$, where each has sensitivity 1, and a threshold $T$. It aims to find the first query (if there is) whose answer is approximately above $T$. See Appendix A for the detailed algorithm. The SVT has been shown to satisfy $\varepsilon$-DP with following utility guarantee.

**Lemma 5** (Extension of Theorem 3.24 in [16]). *With probability at least $1 - \beta$, SVT returns a $k$ such that, for any $i < k$, $f_i(\mathbf{X}) \leqslant T + \frac{6}{\varepsilon} \log(2t/\beta)$, and if $k \neq t + 1$, then $f_k(\mathbf{X}) \geqslant T - \frac{6}{\varepsilon} \log(2t/\beta)$.*

### 3.3 Concentration Inequalities

**Lemma 6** ([26]). *Given $\mathbf{Y} \sim \mathcal{N}(0, \mathbf{I}_{d \times d})$, with probability at least $1 - \beta$,*

$$\|\mathbf{Y}\|_2 \leqslant \eta(d, \beta) := \sqrt{d + 2\sqrt{d \log(1/\beta)} + 2 \log(1/\beta)}.$$

**Lemma 7** ([8, 26]). *Given $\mathbf{W} \sim \mathrm{SGW}(d)$, with probability at least $1 - \beta$,*

$$\|\mathbf{W}\|_2 \leqslant \upsilon(d, \beta) := 2\sqrt{d} + 2d^{1/6} \log^{1/3} d + \frac{6(1 + (\log d/d)^{1/3})\sqrt{\log d}}{\sqrt{\log(1 + (\log d/d)^{1/3})}} + 2\sqrt{2 \log(1/\beta)}.$$

*Also, with probability at least $1 - \beta$,*

$$\|\mathbf{W}\|_F \leqslant \omega(d, \beta) := \sqrt{d^2 + 2\sqrt{d \log(2/\beta)}(1 + \sqrt{2(d-1)}) + 6 \log(2/\beta)}.$$

Ignoring polylogarithmic factors, $\eta(d, \beta)$ and $\upsilon(d, \beta)$ are both in $\tilde{O}(\sqrt{d})$, while $\omega(d, \beta)$ is in $\tilde{O}(d)$. These concentration inequalities are very useful for error analysis. For example, the bound on $\|\mathbf{W}\|_F$ immediately implies that GaussCov has error $\frac{1}{\sqrt{\rho}n} \cdot \omega(d, \beta) = \tilde{O}(d/n)$.

## 4 Trace-sensitive Algorithm

The state-of-the-art trace-sensitive algorithm [2] first obtains an estimate of the eigenvalues, and then iteratively finds the eigenvectors by the exponential mechanism (EM), so we denote this algorithm as EMCov. Under zCDP, it has an error of $\tilde{O}(d^{3/4}\sqrt{\mathrm{tr}}/\sqrt{n} + \sqrt{d}/n)$. Below, we present an algorithm that is simpler, faster, and more accurate, improving the trace-dependent term by a $\sqrt{d}$-factor.

The first step of our algorithm SeparateCov (shown in Algorithm 1) is basically the same as EMCov, where we obtain an estimate of the eigenvalues with half of the privacy budget. [2] uses the Laplace mechanism for pure-DP; for zCDP, we use the Gaussian mechanism, which relies on the $\ell_2$-sensitivity of $\mathbf{\Lambda}$, which we provide in Lemma 10 in the Appendix B. For the eigenvectors, we use GaussCov to obtain a privatized $\widetilde{\mathbf{\Sigma}}_{\mathrm{Gau}}$ with the other half of the privacy budget, and perform an eigendecomposition. Finally, we assemble the eigenvalues of eigenvectors to obtain a privatized $\mathbf{\Sigma}$. It should be clear that, after computing $\mathbf{\Sigma}$, SeparateCov just needs two eigendecompositions and one full matrix multiplication, plus some $O(d^2)$-time operations. On the other hand, EMCov performs $O(d)$ eigendecompositions and matrix multiplications, plus a nontrivial sampling procedure for the EM.

That SeparateCov satisfies $\rho$-zCDP easily follows from the privacy of the Gaussian mechanism and the composition property. The utility is given by the following theorem:

**Theorem 1.** *Given any $\rho > 0$, for any $\mathbf{X} \in \mathcal{B}_d^n$, and any $\beta > 0$, with probability at least $1 - \beta$,* SeparateCov *returns a $\widetilde{\mathbf{\Sigma}}_{\mathrm{Sep}}$ such that $\|\widetilde{\mathbf{\Sigma}}_{\mathrm{Sep}} - \mathbf{\Sigma}\|_F \leqslant \frac{2^{1.25}\sqrt{\mathrm{tr}}}{\rho^{1/4}\sqrt{n}} \cdot \sqrt{\upsilon\left(d, \frac{\beta}{2}\right)} + \frac{\sqrt{2}}{\sqrt{\rho}n} \cdot \eta\left(d, \frac{\beta}{2}\right) = \tilde{O}\left(\frac{d^{1/4}\sqrt{\mathrm{tr}}}{\sqrt{n}} + \frac{\sqrt{d}}{n}\right)$.*

---
**Algorithm 1** SeparateCov
---
**Input:** data $\mathbf{X} \in \mathcal{B}_d^n$; privacy parameter $\rho > 0$.
1: $\mathbf{\Lambda} \leftarrow$ the eigenvalues of $\mathbf{\Sigma} = \frac{1}{n}\mathbf{X}\mathbf{X}^T$
2: $\widetilde{\mathbf{\Lambda}}_{\text{Sep}} \leftarrow \mathbf{\Lambda} + \frac{\sqrt{2}}{\sqrt{\rho}n} \cdot \mathbf{Y}$, where $\mathbf{Y} \sim \mathcal{N}(0, \mathbf{I}_{d \times d})$
3: $\widetilde{\mathbf{\Sigma}}_{\text{Gau}} \leftarrow \text{GaussCov}(\mathbf{X}, \frac{\rho}{2})$
4: $\widetilde{\mathbf{P}}_{\text{Sep}} \leftarrow \mathbf{P}\left[\widetilde{\mathbf{\Sigma}}_{\text{Gau}}\right]$
5: $\widetilde{\mathbf{\Sigma}}_{\text{Sep}} \leftarrow \widetilde{\mathbf{P}}_{\text{Sep}} \widetilde{\mathbf{\Lambda}}_{\text{Sep}} \widetilde{\mathbf{P}}_{\text{Sep}}^T$
6: **return** $\widetilde{\mathbf{\Sigma}}_{\text{Sep}}$
---

**Remark** While SeparateCov strictly improves over EMCov, it does not dominate GaussCov: When $\text{tr} < \tilde{O}(d^{3/2}/n)$, SeparateCov is better; otherwise, GaussCov is better. EMCov is better than GaussCov for a smaller trace range: $\text{tr} < \tilde{O}(\sqrt{d}/n)$.

Theorem 1 implies our worst-case bound by taking $\text{tr} = 1$:

**Theorem 2.** *Given any $\rho > 0$, for any $\mathbf{X} \in \mathcal{B}_d^n$, and any $\beta > 0$, with probability at least $1 - \beta$,* $\text{SeparateCov}$ *returns a* $\widetilde{\mathbf{\Sigma}}_{\text{Sep}}$ *such that* $\|\widetilde{\mathbf{\Sigma}}_{\text{Sep}} - \mathbf{\Sigma}\|_F = \tilde{O}\left(\frac{d^{1/4}}{\sqrt{n}} + \frac{\sqrt{d}}{n}\right)$.

## 5 Tail-sensitive Algorithm

### 5.1 Clipped Covariance

Clipping is a common technique to reduce the sensitivity of functions at the expense of some bias. Given $\tau \geqslant 0$ and a vector $X \in \mathbb{R}^d$, let $\text{Clip}(X, \tau) = \min\left(1, \frac{\tau}{\|X\|_2}\right) \cdot X$. Similarly, for any $\mathbf{X} \in \mathbb{R}^{d \times n}$, $\text{Clip}(\mathbf{X}, \tau)$ denotes the matrix whose columns have been clipped to have norm at most $\tau$. Clipping can be applied to both GaussCov and SeparateCov with a given $\tau$: just run the mechanism on $\frac{1}{\tau} \cdot \text{Clip}(\mathbf{X}, \tau)$ and scale the result back by $\tau^2$. We denote the clipped versions of the two mechanisms as ClipGaussCov and ClipSeparateCov, respectively.

The following lemma bounds the bias caused by clipping in terms of the $\tau$-tail as defined in (1).

**Lemma 8.** $\|\mathbf{\Sigma}(\mathbf{X}) - \mathbf{\Sigma}(\text{Clip}(\mathbf{X}, \tau))\|_F \leqslant \frac{1}{n}\sum_i \left(\|X_i\|_2^2 - \tau^2\right) \cdot \mathbb{I}\left(\|X_i\|_2 \geqslant \tau\right) \leqslant \gamma(\mathbf{X}, \tau)$.

Thus, running the better of ClipGaussCov and ClipSeparateCov yields a total error of $\text{Noise}(\mathbf{X}, \tau, \rho, \beta) + \gamma(\mathbf{X}, \tau)$, where

$$\text{Noise}(\mathbf{X}, \tau, \rho, \beta) = \min\left(\frac{\tau^2}{\sqrt{\rho}n} \cdot \omega(d, \beta), \frac{2^{1.25}\tau\sqrt{\text{tr}}}{\rho^{1/4}\sqrt{n}} \cdot \sqrt{\upsilon\left(d, \frac{\beta}{2}\right)} + \frac{\sqrt{2}\tau^2}{\sqrt{\rho}n} \cdot \eta\left(d, \frac{\beta}{2}\right)\right), \quad (4)$$

which is the exact version of (2). Note that the trace-sensitive term is only scaled by $\tau$, which follows from the proof of Theorem 1 when all $X_i$ live in $\tau \cdot \mathcal{B}_d$.

Ideally, we would like to find the optimal noise-bias trade-off, i.e., achieving an error of $\min_\tau(\text{Noise}(\mathbf{X}, \tau) + \gamma(\mathbf{X}, \tau))$. Two issues need to be addressed towards this goal: The first, minor, issue is that $\text{tr}$ is sensitive, so we cannot use it directly to decide whether to use ClipGaussCov or ClipSeparateCov. This can be addressed by using a privatized upper bound of $\text{tr}$. The more challenging problem is how to find the optimal $\tau$ in a DP fashion. This problem has been well studied for the clipped mean estimator [18, 3], where it can be shown that setting $\tau$ to be the $\tilde{O}(\sqrt{d})$-th largest $\|X_i\|_2$ results in the optimal noise-bias trade-off [18]. Then the problem boils down to finding a privatized quantile, for which multiple solutions exist [18, 12, 32, 6, 37]. For the clipped mean estimator, using such a quantile of the norms results in the optimal trade-off because $\text{Noise}(\mathbf{X}, \tau)$ takes the simple form $\tilde{O}(\tau\sqrt{d}/n)$. In fact, if we only had ClipGaussCov, setting $\tau$ to be the $\tilde{O}(d)$-th largest $\|X_i\|_2$ would also yield an optimal trade-off, as ClipGaussCov is really just clipped mean in $d^2$ dimensions. However, due to the trace-sensitive noise term, it is no longer the case. In Appendix F, we give examples showing that no quantile, whose rank may arbitrarily depend on $d, n, \text{tr}$, can achieve

an optimal trade-off even ignoring polylogarithmic factors. It thus calls for a new threshold-finding mechanism, which we describe next.

## 5.2 Adaptive Covariance: Finding the Optimal Clipping Threshold

Our basic idea is to try successively smaller values $\tau = 1, \frac{1}{2}, \frac{1}{4}, \ldots$. As we reduce $\tau$, the noise decreases while the bias increases. We should stop when they are approximately balanced, which would yield a near-optimal $\tau$.

To do so in a DP manner, we need to quantify the noise and bias. Consider the bias first. Given a $\tau$, we divide the interval $(\tau, 1]$ into sub-intervals $(\tau, 2\tau], (2\tau, 4\tau], \ldots, (\frac{1}{2}, 1]$. For any $X \in \mathbf{X}$ such that $\|X\|_2 \in (2^s, 2^{s+1}]$, let $\check{X} = \mathrm{Clip}(X, \tau)$ and then by Lemma 8,

$$\|XX^T - \check{X}\check{X}^T\|_F \leqslant 2^{2s+2} - \tau^2. \tag{5}$$

That is, clipping $X$ can at most lead to $\frac{1}{n} \cdot (2^{2s+2} - \tau^2)$ bias. Besides, since $\|X\|_2 \in (2^s, 2^{s+1}]$, we have

$$2^{2s+2} - \tau^2 \leqslant 2^{2s+2} \leqslant 2 \cdot \|X\|_2^2. \tag{6}$$

Then, given $\mathbf{X}$, for any $s \in \mathbb{Z}$, we define $\mathrm{Count}_s(\mathbf{X}) := \left|\left\{X_i : \|X_i\|_2 \in (2^s, 2^{s+1}]\right\}\right|$. It is easy to see for any $\mathbf{X} \sim \mathbf{X}'$, $\mathrm{Count}_s$ differs by at most 1, so does the sum of any subset of $\mathrm{Count}_s$'s. We can define an upper bound on the bias: $\widehat{\mathrm{Bias}}(\mathbf{X}, \tau) := \frac{1}{n} \cdot \sum_{s=\log_2(\tau)}^{s<0} \mathrm{Count}_s \cdot (2^{2s+2} - \tau^2)$. Let $\check{\mathbf{X}} = \mathrm{Clip}(\mathbf{X}, \tau)$. By (5) and (6), we have

$$\frac{1}{n}\|\mathbf{X}\mathbf{X}^T - \check{\mathbf{X}}\check{\mathbf{X}}^T\| \leqslant \widehat{\mathrm{Bias}}(\mathbf{X}, \tau) \leqslant 2 \cdot \gamma(\mathbf{X}, \tau). \tag{7}$$

By the property of $\mathrm{Count}_s$'s, given any $\tau$, the sensitivity of $\widehat{\mathrm{Bias}}(\cdot, \tau)$ is bounded by $\frac{1}{n}$.

Now we turn to the noise. Recall that $\mathrm{Noise}(\mathbf{X}, \tau, \rho, \beta)$ is the smaller of two parts. The first part $\mathrm{GaussNoise}(\tau, \rho, \beta) := \tau^2 \cdot \frac{1}{\sqrt{\rho}n} \cdot \omega(d, \beta)$ is independent of $\mathbf{X}$, so can be used directly. The second part depends on $\mathrm{tr}$, is thus sensitive. Since its sensitivity is $\frac{1}{n}$, we can easily privatize it by adding a Gaussian noise of scale $\Theta\left(\frac{1}{\sqrt{\rho}n}\right)$. For technical reasons, we need to use an upper bound, so we add $\Theta\left(\frac{\log(1/\beta)}{\sqrt{\rho}n}\right)$ to it so as to obtain a privatized $\widehat{\mathrm{tr}} \geqslant \mathrm{tr}$. Then we set

$$\mathrm{SeparateNoise}(\widehat{\mathrm{tr}}, \tau, \rho, \beta) := \tau \cdot \frac{2^{1.25}\sqrt{\widehat{\mathrm{tr}}}}{\rho^{1/4}\sqrt{n}} \cdot \sqrt{\upsilon\left(d, \frac{\beta}{2}\right)} + \tau^2 \cdot \frac{\sqrt{2}}{\sqrt{\rho}n} \cdot \eta\left(d, \frac{\beta}{2}\right),$$

and use $\widehat{\mathrm{Noise}}(\widehat{\mathrm{tr}}, \tau, \rho, \beta) := \min\left(\mathrm{GaussNoise}(\tau, \rho, \beta), \mathrm{SeparateNoise}(\widehat{\mathrm{tr}}, \tau, \rho, \beta)\right)$ as a DP upper bound of $\mathrm{Noise}(\mathbf{X}, \tau, \rho, \beta)$. Note that given $\widehat{\mathrm{tr}}$, $\widehat{\mathrm{Noise}}(\widehat{\mathrm{tr}}, \tau, \rho, \beta)$ is independent of $\mathbf{X}$.

Finally, we run SVT on the following sequence of sensitivity-1 queries with $T = 0$:

$$\mathrm{Diff}(\mathbf{X}, \widehat{\mathrm{tr}}, \tau, \rho, \beta) := n \cdot \left(\widehat{\mathrm{Bias}}(\mathbf{X}, \tau) - \widehat{\mathrm{Noise}}(\widehat{\mathrm{tr}}, \tau, \rho, \beta)\right), \tau = 1, \frac{1}{2}, \ldots, 2^{-dn}.$$

The SVT would return a $\tau$ that balances the bias and noise. After finding such a $\tau$, we choose to run either GaussCov or SeparateCov by comparing $\mathrm{GaussNoise}(\tau, \rho, \beta)$ and $\mathrm{SeparateNoise}(\widehat{\mathrm{tr}}, \tau, \rho, \beta)$. As the sequence consists of $dn$ queries, SVT has an error of $O(\log(dn))$, which, as we will show, affects the optimality by a logarithmic factor. Meanwhile, the smallest $\tau$ we search over will induce an additive $2^{-dn}$ error.

The algorithm above can almost give us the desired error bound in (3), except that one thing may go wrong: The SVT introduces an error that is a logarithmic factor larger than the optimum, but at least $\widetilde{\Omega}(1/n)$. This would be fine as long as there is one $X_i$ with $\|X_i\|_2 \geqslant \widetilde{\Omega}(1)$, so that the optimum error is $\widetilde{\Omega}(1/n)$. However, when all the $X_i$'s have very small norms, say $1/n^2$, the $\widetilde{\Omega}(1/n)$ error from SVT would not preserve optimality. To address this issue, we first find the radius $\mathrm{rad}(\mathbf{X}) = \max_i \|X_i\|_2$, and use it to clip $\mathbf{X}$. The following lemma shows that, under DP, it is possible to find a 2-approximation of $\mathrm{rad}(\mathbf{X})$ plus an additive $b$ so that only $O(\log\log(1/b))$ vectors are clipped. This allows us to set $b = 2^{-dn}$ while only incurring an $O(\log dn)$ error. Nicely, they match the additive and multiplicative errors that already exist from the SVT, so there is no asymptotic degradation in the optimality.

**Algorithm 2** AdaptiveCov

---

**Input:** data $\mathbf{X} \in \mathcal{B}_d^n$; privacy parameter $\rho > 0$; high probablity parameter $\beta$.

1: $\tilde{r} \leftarrow \text{PrivRadius}(\mathbf{X}, \frac{\sqrt{\rho}}{2}, \frac{\beta}{8}, 2^{-2dn})$

2: $\widetilde{\mathbf{X}} \leftarrow \text{Clip}(\mathbf{X}, \tilde{r})$

3: $\widetilde{\text{tr}} \leftarrow \frac{1}{n} \sum_i \|\widetilde{X}_i\|_2^2$

4: $\widehat{\text{tr}} \leftarrow \min\left(\widetilde{\text{tr}} + \frac{2\tilde{r}^2}{\sqrt{\rho}n} \cdot \mathcal{N}(0,1) + \frac{2\sqrt{2}\tilde{r}^2}{\sqrt{\rho}n} \cdot \sqrt{\log(8/\beta)}, \ \tilde{r}^2\right)$

5: $\tilde{t} \leftarrow \log_2(\tilde{r})+1-\text{SVT}\left(\left\{\text{Diff}\left(\widetilde{\mathbf{X}}, \widehat{\text{tr}}, \tilde{r}, \frac{\rho}{2}, \frac{\beta}{2}\right), \text{Diff}\left((\widetilde{\mathbf{X}}, \widehat{\text{tr}}, \frac{\tilde{r}}{2}, \frac{\rho}{2}, \frac{\beta}{2}\right), \dots, \text{Diff}\left((\widetilde{\mathbf{X}}, \widehat{\text{tr}}, 2^{-dn}, \frac{\rho}{2}, \frac{\beta}{2}\right)\right\}, 0, \frac{\sqrt{\rho}}{\sqrt{2}}\right)$

6: $\tilde{\tau} \leftarrow \min\left(2^{\tilde{t}+1}, \tilde{r}\right)$

7: **if** $\text{SeparateNoise}(\widehat{\text{tr}}, \tilde{\tau}, \frac{\rho}{2}, \frac{\beta}{2}) \geqslant \text{GaussNoise}(\tilde{\tau}, \frac{\rho}{2}, \frac{\beta}{2})$

8: $\quad \widetilde{\mathbf{\Sigma}}_{\text{Ada}} \leftarrow \text{ClipGaussCov}(\widetilde{\mathbf{X}}, \frac{\rho}{2}, \tilde{\tau})$

9: **else**

10: $\quad \widetilde{\mathbf{\Sigma}}_{\text{Ada}} \leftarrow \text{ClipSeparateCov}(\widetilde{\mathbf{X}}, \frac{\rho}{2}, \tilde{\tau})$

11: **return** $\widetilde{\mathbf{\Sigma}}_{\text{Ada}}$

---

**Lemma 9** ([12]). *For any $\varepsilon > 0$, $\beta > 0$ and $b > 0$, given $\mathbf{X} \in \mathcal{B}_d^n$, with probability at least $1 - \beta$, $\texttt{PrivRadius}$ returns a $\tilde{r} = \texttt{PrivRadius}(\mathbf{X}, \varepsilon, \beta, b)$ such that $\tilde{r} \leqslant 2 \cdot \text{rad}(\mathbf{X}) + b$ and $|\{\|X_i\|_2 > \tilde{r}\}| = O\left(\frac{1}{\varepsilon} \log \frac{\log(\text{rad}(\mathbf{X})/b)}{\beta}\right)$.*

The complete algorithm is given in Algorithm 2. Its privacy follows from the privacy of $\texttt{PrivRadius}$, SVT, GaussCov, SeparateCov, and the composition theorem of zCDP; its utility is analyzed in the following theorem:

**Theorem 3.** *Given any $\rho > 0$ and $\beta > 0$, for any $\mathbf{X} \in \mathcal{B}_d^n$, with probability at least $1 - \beta$, AdaptiveCov returns a $\widetilde{\mathbf{\Sigma}}_{\text{Ada}}$ such that*

$$\left\|\widetilde{\mathbf{\Sigma}}_{\text{Ada}} - \mathbf{\Sigma}\right\|_F = O\left(\min_\tau\left(\text{Noise}\left(\mathbf{X}, \tau, \frac{\rho}{2}, \frac{\beta}{2}\right) \cdot \frac{\log(1/\beta)^{1/4}}{\rho^{1/4}} \cdot \log(n) + \gamma(\mathbf{X}, \tau) \cdot \frac{\log(dn/\beta)}{\sqrt{\rho}}\right) + 2^{-dn}\right)$$

$$= \tilde{O}\left(\min_\tau\left(\text{Noise}\left(\mathbf{X}, \tau\right) + \gamma(\mathbf{X}, \tau)\right) + 2^{-dn}\right).$$

## 6 Experiments

We conducted experiments[5] to evaluate our algorithms on both synthetic and real-world datasets. We compare SeparateCov and AdaptiveCov against GaussCov [17], EMCov [2]. We implemented EMCov in Python following the pseudo-code provided in [2] and the descriptions of the sampling algorithm in [24]. We also tested CoinPress [8], but since it is designed to minimize the Mahalanobis error, it does not perform well when measured in Frobenius error. The two distance measures coincide when $\mathbf{\Sigma}$ is well-conditioned but in this case, CoinPress degenerates into GaussCov. Therefore, we omit it from the reported results. As a baseline, we include returning a zero matrix, which has error $O(\text{tr})$, hence a trivial trace-sensitive algorithm. When $\text{rad}(\mathbf{X})$ is much smaller than 1, it is unfair for GaussCov and EMCov, so we scale all datasets such that $0.5 \leqslant \text{rad}(\mathbf{X}) \leqslant 1$. As a result, we do not need the step to obtain a private radius in AdaptiveCov, either. Each experiment is repeated 50 times, and we report the average error. Furthermore, we have also conducted experiments under pure-DP; the results can be found in Appendix J.

### 6.1 Synthetic Datasets

We generate synthetic datasets by first following the method in [2], to obtain a matrix $\mathbf{X} = \mathbf{ZU}$, where $\mathbf{U} \in \mathbb{R}^{d \times d}$ is sampled from $U(0, 1)$, and $\mathbf{Z} \in \mathbb{R}^{n \times d}$ is sampled from $\mathcal{N}(0, \mathbf{I})$. Then the vectors in $\mathbf{X}$ are adjusted to be centred at 0 and their norms scaled. In [2], the vectors are scaled to have unit $\ell_2$ norm; in our experiments, to better control $\text{tr}$ and data skewness, we scale the norms so that they

---

[5]The code can be found at `https://github.com/hkustDB/PrivateCovariance`.

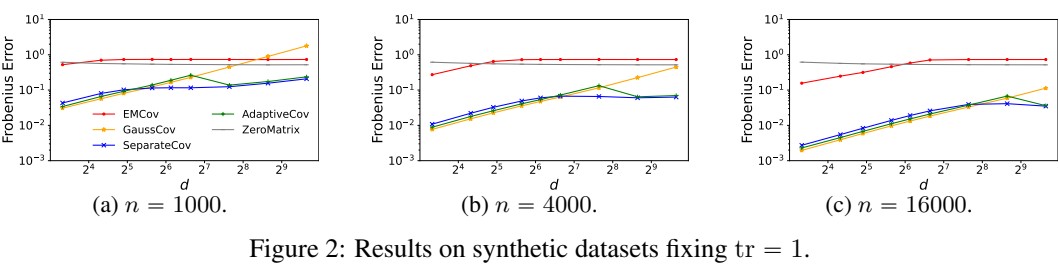

Figure 2: Results on synthetic datasets fixing $\mathrm{tr} = 1$.

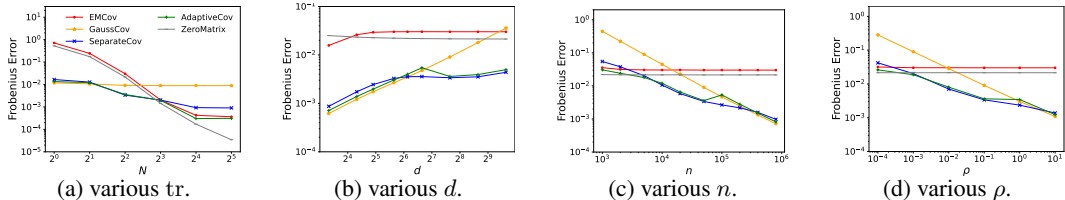

Figure 3: Results on synthetic datasets as $d, n, N$ or $\rho$ varies.

follow the Zipf's law. More precisely, we divide the norms into $N$ bins. The number of vectors in the $k$-th bin is proportional to $1/k^s$ and their norm is $2^{k-N}$. The parameter $s$ characterizes the skewness, which we fix as $s = 3$. Note that $N = 1$ corresponds to the unit-norm case with $\mathrm{tr} = 1$.

The results on $\mathrm{tr} = 1$ case are shown in Figure 2, which correspond to the worst-case bounds. The $\rho$ here is fixed at $0.1$ and we examine the error growth w.r.t. $d$ for $n = 1000, 4000, 16000$. The results generally agree with the theory: For low $d$, $\mathrm{GaussCov}$ is (slightly) better than $\mathrm{SeparateCov}$, while the latter is much better for high $d$. $\mathrm{AdaptiveCov}$ is able to choose the better of the two adaptively, with a small cost due to allocating some privacy budget to estimate $\mathrm{tr}$. Actually, if $\mathrm{AdaptiveCov}$ is given the precondition that all norms are $1$, this extra cost can be saved.

Next, we vary one of the parameters while fixing the others at their default values $d = 200$, $n = 50000$, $N = 4$ and $\rho = 0.1$, and the results are reported in Figure 3. The most interesting case is Figure 3(a), where we increase $N$, hence reducing $\mathrm{tr}$, which demonstrates the trace-sensitive bounds. Clearly, $\mathrm{GaussCov}$ is not trace-sensitive, while the other 4 methods are. In fact, returning the zero matrix is the best trace-sensitive algorithm if $\mathrm{tr}$ is sufficiently small. However, this may not be very meaningful in practice, as $N = 2^5$ means that most data have norm $2^{-31}$ but a few have norm $1$. These few may be outliers and should be removed anyway. Figure 3(b)–(d) shows that higher $d$, smaller $n$, and smaller $\rho$ all have similar effects, i.e., $\mathrm{SeparateCov}$ becomes better while $\mathrm{GaussCov}$ becomes worse, while $\mathrm{AdaptiveCov}$ is able to pick the better one.

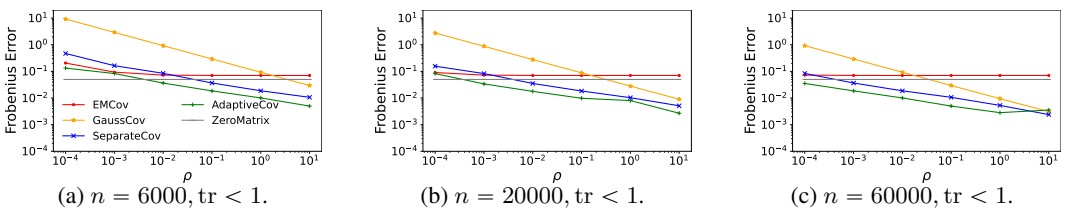

Figure 4: Results on MNIST dataset.

## 6.2 Real-world Datasets

We also evaluate the algorithms on two real-world datasets. The first dataset is the MNIST [27] dataset, which contains images of handwritten digits. We use its training dataset which contains $60,000$ images represented as vectors in $\mathbb{Z}_{255}^d$, where $d = 784 = 28 \times 28$. These vectors are normalized by $255\sqrt{d}$ in the experiments. We estimate $\widetilde{\Sigma}$ using samples containing all the digits, we also estimate $\widetilde{\Sigma}$ corresponding to individual digits (reported in the Appendix J). In the first case, $\widetilde{\Sigma}$ can be used for further dimensionality reduction analysis; in the second case, individual $\widetilde{\Sigma}$ can be used for modelling the distributions of individual digits, which together can be used in a collective

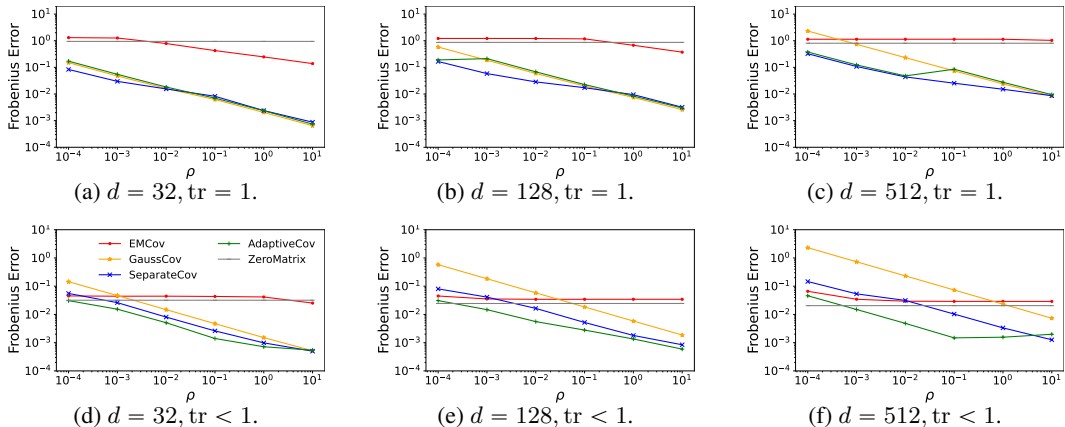

Figure 5: Results on the news dataset.

model for classification (e.g. using a mixture model). The second dataset contains news commentary data [33] consisting of approximately $15,000$ articles, each containing $500 - 4300$ words, which we convert into vectors of various dimensions using the hashing trick implemented in the scikit-learn package. In this case, the estimated $\widetilde{\Sigma}$ can be used to help with further feature selection for NLP models, for example. These vectors are normalized to have unit $\ell_2$ norm or normalized by the max $\ell_2$ norm.

The experimental results on these two real dataset are shown in Figure 4 and 5, where we vary $n, d$, and $\rho$, respectively. On these results, we see that $\mathrm{GaussCov}$ never outperforms $\mathrm{SeparateCov}$, except for a very small advantage in a few cases where we have $\mathrm{tr} = 1$, a low $d$, and a high $\rho$. Another interesting observation is that $\mathrm{AdaptiveCov}$ outperforms both $\mathrm{GaussCov}$ and $\mathrm{SeparateCov}$ in many cases, something that is not apparent on the synthetic datasets. We believe that this is because these real datasets have heavier tails than the Zipf distribution (we used $s = 3$ for Zipf), which makes the adaptive clipping threshold selection more effective. This really demonstrates the benefits of a tail-sensitive bound.

## Acknowledgements

This work has been supported by HKRGC under grants 16201819, 16205420, and 16205422. We would like to thank Aleksandar Nikolov for helpful discussions on the projection mechanism and the anonymous reviewers who have made valuable suggestions on improving the presentation of the paper.

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
