# OpenReview forum: "Differentially Private Covariance Revisited"
_NeurIPS.cc/2022/Conference — NeurIPS 2022 Accept_

### Official Review · Reviewer_vJW1 · 2022-06-24

**Rating:** 7
**Confidence:** 4
**Soundness:** 3 good
**Presentation:** 3 good
**Contribution:** 3 good

**Summary:**

The authors consider the problem of releasing the empirical covariance matrix under concentrated differential privacy. Assuming a boundedness assumption on the l2 norms of the data points, they propose a new algorithm for the problem and show theoretically that it has a better error bound compared to the previous work in the regime where d is very large. They also give a trace-sensitive guarantee to exploit the situations where an average point has a small norm, as well as a tail-sensitive guarantee.

The first algorithm is a combination of learning the eigenvalues and running PCA. I believe both of these individual steps are known in the literature. The tail sensitive result has more subtleties in order to pick the right threshold to clip the further points.

**Questions:**

+ It is somewhat strange that in some experiments the Frobenius error increases with n. Can you elaborate on that?

+ How does your method compare to the previous work in terms of dependence on the privacy parameters?

+ How did you pick epsilon in the experiments?



**Strengths And Weaknesses:**

Strength:

+ contributions to a well-defined and well-motivated theoretical problem in the DP literature
+ simple and intuitive algorithm which is implemented by the authors
+ empirical results demonstrating the effectiveness of the method

Weaknesses:

+ The problem of mean estimation has been mentioned throughout the text many times but the exact definition is missing. Do you mean estimation w.r.t. l2 error? It will help to define the problem of mean estimation and even covariance estimation formally.
+ The error bounds are only stated in terms of dimension and number of samples. The dependence on the privacy parameter is not explicit in these results
+ As far as I can tell, the experiments results also don't mention the value of epsilon either
+ It is assumed that all data points lie in a unit l2 ball. If we relax this to a ball of radius R, then I believe the authors error bound grows linearly with R. It is not clear if this linear dependence is optimal. Some discussion of the dependence on R will be useful.
+ in some experiments the error increases with the number of samples, which seems odd.

---

> ### Author Response · Authors · 2022-07-30
> **Reply to Reviewer vJW1**
>
> Weakness 1: Yes, we refer to mean estimation with l2 error and we will clarify it.
>
> Weakness 2: The dependency on the privacy parameters is explicitly given in the formal theorems, but omitted in the introduction for brevity.
>
> Weakness 3/Question 3: We use zCDP, whose parameter is $\rho$ (we give the values of $\rho$).  As mentioned in the experiment section, this can be converted to $(\varepsilon,\delta)$-DP via standard formulas in line 122-123.
>
> Weakness 4: Yes, the error has a linear dependency on $R^2$ and this is optimal.  When data is scaled by $R$, covariance is scaled by $R^2$.  Suppose one can achieve a lower dependency, say $R^{1.5}$.  Then, given a dataset in the unit ball, we scale it up to $R$, apply this algorithm, and scale the result down.  This would improve the result for the original dataset by a factor $R^{-0.5}$.
>
> Weakness 5/Question 1: Thank you for your careful review! This is a good observation and is because AdaptiveCov chooses the better of GaussCov and SeparateCov, but this decision has to be done privately, thus incurring a constant-factor gap.  For a few datasets, AdaptiveCov may choose the worse one (up to a constant factor).
>
> Question 2: All GaussCov, EMCov, SeparateCov, and AdaptiveCov have the $\varepsilon^{-1}$ ($\rho^{-1/2}$) dependency.
>
> Question 3: The values for $\rho$ we use are in the range [1e-4,1e1], which covers the range [0.005, 0.5] used by CoinPress (NeurIPS'20), and roughly covers the range implied by the $\varepsilon$'s used in experiments of EMCov (NeurIPS'19).

---

### Official Review · Reviewer_1ufN · 2022-07-08

**Rating:** 8
**Confidence:** 3
**Soundness:** 4 excellent
**Presentation:** 4 excellent
**Contribution:** 3 good

**Summary:**

The paper gives improved error bounds for releasing an approximation of a covariance matrix $\Sigma$ under approximate differential privacy/zCDP, breaking lower bounds that hold under pure DP for dimension d between n^2/3 and n^2, where n is the number of data points. (Privacy is relative to changing one data point, a d-dimensional *unit vector*.) The new bounds improve both existing worst-case bounds and bounds expressed in terms of the trace of the covariance matrix (generalizing the worst-case bounds) measured in terms of *Frobenius norm* of the error.

The approach is surprisingly simple, and also more computationally efficient than previous work: Release the eigenvalues with DP by adding noise to the exact eigenvalues. Estimate the eigenvectors by doing an eigendecomposition of the covariance matrix made private using the Gaussian mechanism. Then put everything together. (The paper also has a more sophisticated approach that improves on the simple method in a data-dependent way by carefully clipping data.)

Practicality of the methods is investigated through experimental comparison with the Gaussian mechanism and EMCov (NeurIPS '19). Good improvements are demonstrated on synthetic data as well as two real-world datasets.

**Questions:**

* You assume vectors are unit length --- can you say something about to what extent this is a limitation of the approach, or whether (say) you can generalize to vectors with lengths in some fixed range?
* In the case of vectors from {$-1/\sqrt{d}, 1/\sqrt{d}$}$^d$ one can apply results on release of 2-way marginals to estimate the covariance matrix. How do your results compare to that approach?
* Do you have a use case/justification why one might be interested in the covariance matrix for the data sets you study? If not, are there other data sets with a more compelling use case you could try?

Comments
* It would be good to describe GaussCov together with Algorithm 1, rather than hidden in the preliminaries
* Line 263: Do you mean $10^{-10}$ rather than $1e^{-10}$?

**Limitations:**

I am satisfied with the paper discussion of limitations. Since this is a very general question it seems unlikely that there are negative societal impacts (and the authors do not discuss this).

**Strengths And Weaknesses:**

Strengths
* A polynomial improvement in error (for sufficiently large dimension) compared to EMCov (NeurIPS '19) as well as the Gaussian mechanism
* The mechanism is very neat --- it is surprising that it pays off to estimate eigenvalues and eigenvectors separately in different ways
* Experimental evidence of practicality

Weaknesses
* It is not clear how the bounds achieved relate to what can be achieved using general query release methods, for example simhash combined with methods for release of 2-way marginals
* There is no experimental comparison to CoinPress (NeurIPS '20), since CoinPress optimizes a Manahalobis distance and in general performs poorly in terms of Frobenius error; however there are covariance matrices for which these distance measures coincide, and where a comparison would make sense
* Arguably, for the real-world data sets used in experiments it does not seem natural to be interested in computing the covariance matrix

---

> ### Author Response · Authors · 2022-07-30
> **Reply to Reviewer 1ufN**
>
> Weakness 1/Question 2: Thank you for mentioning that the 2-way marginal problem is a special case of the covariance problem where the input vectors are from $\\{-1/\sqrt{d},1/\sqrt{d}\\}^d$. The best result for this problem achieves an l2 error of $\tilde{O}(\min(d/n, d^{1/4}/\sqrt{n}))$ [b], which matches our worst-case result (i.e., the better of GaussCov and SeparateCov).  The differences are that (1) their algorithm cannot handle vectors from the unit ball; (2) [b] has the computational complexity $O(nd^3)$ while ours is $O(nd^2+d^3)$; (3) there are no \``trace-sensitive\'' or \``tail-sensitive\'' algorithms for the 2-way marginal problem as all input vectors have the same norm.  We should definitely include this comparison to the paper.
>
> Weakness 2: The two distance measures coincide when $\mathbf{\Sigma}$ is well-conditioned, but in this case, CoinPress degenerates into GaussCov.  For other cases, we do have the experimental results for CoinPress as well, but just omitted them from the paper as it is significantly worse than the other methods.
>
> Weakness 3/Question 3: For evaluation of the algorithms, we choose two datasets that are different in nature (one containing image data, the other containing news data). We briefly described the potential use cases in line 298-305.
>
> Question 1: Please note that we assume the vectors are in the unit ball, not necessarily unit length. And for vectors with lengths in some fixed range $R$, they can be scaled down to have a length at most $1$ and the error will depend on $R^2$.
>
> Comment 2: Sorry for the typo, we mean the scientific notation 1e-10, which is equal to $10^{-10}$. Thanks for pointing it out!
>
> [b] Dwork, C., Nikolov, A., & Talwar, K. (2015). Efficient algorithms for privately releasing marginals via convex relaxations. Discrete & Computational Geometry, 53(3), 650-673.

---

> > ### Comment · Reviewer_1ufN · 2022-08-08
> > **Comment on author reply**
> >
> > Thanks for the clarifications!
> >
> > Regarding question 2, it is unclear to me how much generality is lost by considering vectors from {$-1/\sqrt{d}, 1/\sqrt{d}$}$^d$. After all, we can use simhash to construct vectors in {$-1/\sqrt{d}, 1/\sqrt{d}$}$^d$ whose dot products are in a 1-1 correspondence with the dot products of the original, normalized vectors, so possibly there is an efficient reduction of the general case to 2-way marginals. Though the direct approach presented in this paper may be preferable, it would be good to discuss this alternative approach.
> >
> > It makes sense to not include results against CoinPress, but again it would be good to discuss this a bit more.

---

> > > ### Author Response · Authors · 2022-08-08
> > > **Reply to Reviewer 1ufN**
> > >
> > > Thank you for your helpful comments!
> > >
> > > Question 2: This is an interesting observation! Yes, we can use simhash to transform a vector from the unit sphere to one in $\\{-1/\sqrt{d},1/\sqrt{d}\\}^d$ and then apply the mechanism for the 2-way marginal problem. That does work but it is unclear what the error is: covariance is computed from the cross products of the vectors and it seems unclear how much their cross products differ since simhash does some ''rounding''. We will add this discussion in our paper.
> > >
> > > Weakness 2: We agree. We will discuss this a bit more and include the discussion from our previous response.
> > >
> > > Any further comments are welcomed!

---

### Official Review · Reviewer_pHxC · 2022-07-10

**Rating:** 7
**Confidence:** 3
**Soundness:** 4 excellent
**Presentation:** 1 poor
**Contribution:** 3 good

**Summary:**

This paper revisits the problem of differentially private (DP) covariance estimation. It provides bounds for three settings of the problem: (1) worst-case setting, (2) trace-sensitive setting, and (3) tail-sensitive setting. Several improvements are made for certain parameter regimes and types of DP notions over the prior work on this problem, and the algorithms provided are computationally efficient.

In the worst-case setting, in the high-dimensional case ($d>n^{2/3}$), a separation between the bounds is shown between pure DP and zCDP.

In the trace-sensitive setting, improvements in both pure DP and zCDP are shown via algorithms that are also more computationally efficient than those of the prior work.

The tail-sensitive setting also sees some improvements.

Both theoretical and experimental results are provided in this work, with the latter showing improvements on both synthetic and real-world datasets.

Edit: Updated my score.

**Questions:**

I do have a few questions.

1. Where are the results for the worst-case setting? I would expect a theorem or something in the main body or somewhere about it. I read a subsection about it in the introduction, but that's pretty much it.
2. Section 1.2, Line 39. What is $\Sigma$? Are you defining a notation here, or referencing something else?
3. Line 41. Is $A = (X_1,\dots,X_d)$? Or is $A$ the normalised, empirical covariance matrix? Please, be clear about your notations! It's not the responsibility of the reader to infer them.

**Limitations:**

1. Please, explain what you mean by things like "worst-case", "trace-sensitive", and "tail-sensitive". These terms don't mean anything by themselves, as far as I know. It may be obvious to the authors, but even a one-liner for each in the main body would make it so much easier to distinguish among these settings. It is a little unpleasant as a reviewer and a reader, to be getting into the manuscript without knowing what the context is.
2. The first part of Lines 29-30 is a bit misleading. The lower bound from [KLSU19] is under a distributional assumption of a Gaussian with an almost identity covariance. The setting in this paper seems more empirical and under bounded norm assumption for the data.
3. In the second half of Section 1.3 (mostly from Line 75 onward), things don't make a lot of sense here without any context about what these settings are. The results are just stated without any meaning.
4. In the first paragraph of Section 2, there are really broad statements about the prior work without stating any specific error metrics and range bounds for the data. I would be more specific here.
5. In the related work about low-rank approximations under DP, I would also state the recent work of Singhal-Steinke 2021 on private subspace estimation.
6. Also, for the citation of DP with Definition 1, along with [DR14], please use [DMNS06]. The latter is the most important citation for the subject.

**Strengths And Weaknesses:**

Strengths:
1. Improvements are made upon the prior work in all the three listed settings, which I would say is important. Polynomial improvements in the results over prior work in the listed parameter regimes look quite decent, I think.
2. The separation in the worst-case setting for the high-dimensional setting between pure DP and zCDP regimes is somewhat significant. It means that for a decent range of parameters, there are newer and better tools at our disposal. I'm not fully convinced though that this regime is the most important one in terms of the significance of the contributions.
3. The algorithms (SeparateCov and AdaptiveCov) look simple enough, and easy to understand. It is surprising that such simple algorithms are able to improve upon existing work.
4. The experimental results are also in favour of their work for both synthetic and real-world datasets (MNIST and news commentary). In each, different variations are considered according to different parameters like $n$, $tr$, $d$, and $\rho$. AdaptiveCov seems more consistent in the experiments on the latter maybe because it seems to be choosing the best of (truncated) SeparateCov and GaussCov.

Weaknesses:
1. My main complaint with this work is the writing quality. In many places, the notations are unclear, broad statements are made, and things are ill-defined. I will elaborate all of these in later sections of this review.
2. I'm a little concerned that in the trace-sensitive setting, the SeparateCov algorithm is not able to improve upon the existing work for all settings. For lower trace, SeparateCov is better than GaussCov, but otherwise, it is not. So, it seems to be of limited utility to some extent.
3. The above seems to be a general concern for me for the entire paper, to be honest. Yes, for a decent range, the worst-case bounds are nicer and give a separation between two forms of DP. I'm not sure how strong the results are owing to these constraints on the dimension.

---

> ### Author Response · Authors · 2022-07-30
> **Reply to Reviewer pHxC**
>
> Weakness 2&3: We note a very recent lower bound of $\tilde{\Omega}(d/n)$ for $d<\sqrt{n}$ under $(\varepsilon, \delta)$-DP [a].  Thus, in the high-trace (i.e., worst-case) setting, GaussCov is already optimal for $d<\sqrt{n}$, while we improve the case $d>n^{2/3}$. Whether one can do better than GaussCov for $\sqrt{n}<d<n^{2/3}$ remains open.
>
> Question 1: Our worst-case bound follows from the trace-sensitive result by taking $\mathrm{tr}=1$.  Please see line 51-52.
>
> Question 2: We sometimes simply write $\mathbf{\Sigma}(\mathbf{X})$ as $\mathbf{\Sigma}$.  Sorry about the confusion!
>
> Question 3: $\mathbf{A}$ is just the input to the functions $\mathbf{P}[\cdot]$ and $\mathbf{\Lambda}[\cdot]$.  We often take $\mathbf{A}=\mathbf{\Sigma} = \mathbf{\Sigma}(\mathbf{X})$, in which case we simplify the notation to $\mathbf{P}$ and $\mathbf{\Lambda}$.
>
> Limitation 1: That depends on the parameters used in the error bound. Worst-case bounds only use $d$ and $n$; trace-sensitive bounds also use $\mathrm{tr}$; tail-sensitive bounds also use $\gamma$.
>
> Limitation 2: Statistical lower bounds imply empirical lower bounds via a standard reduction; see e.g. [16].
>
> Limitation 3: It should be obvious that the tail-sensitive bound is always no worse than the trace-sensitive bound.  The example in line 75-80 is meant to show that the tail-sensitive bound is asymptotically better than the trace-sensitive bound for certain distributions.
>
> Limitation 4: These prior results have the same error metric as our worst-case result and we will explicitly mention this.
>
> Limitation 5&6: Sure!
>
> We will make the modifications mentioned above and thanks!
>
> [a] Kamath, G., Mouzakis, A., & Singhal, V. (2022). New Lower Bounds for Private Estimation and a Generalized Fingerprinting Lemma. arXiv preprint arXiv:2205.08532.

---

> > ### Comment · Reviewer_pHxC · 2022-08-06
> > **Response to the Authors**
> >
> > Thank you for your attempt to provide the clarifications!
> >
> > Regarding question 1: I understand the math that setting $tr=1$ would do the trick for the worst-case scenario. My point is that an explicit theorem/corollary about it somewhere would be nice because that's one of your main results. It's more of a question about your presentation than about the result itself.
> >
> > Regarding limitation 1: I don't think my point was clear here. I'm essentially asking the authors to define the terms "worst-case", "trace-sensitive", and "tail-sensitive" in the main body of the paper **explicitly**. What do those settings mean? Sure, I can read the math eventually, and figure out what is going on, but it is generally not a good idea to force the reader to search around in your paper for their meanings, especially when there is not explicit definition. Again, it is a comment about the presentation, and not about the results themselves. I hope that's clear now!
> >
> > Regarding limitation 2: I think what I'm saying is again not clear. The setting you have in your paper is where the rows (or the trace) have bounded norm of $1$. [KLSU19] does not have that assumption -- their rows (when scaling down the range-bounds on the covariance matrix of the Gaussian) will have similar, *high-probability* bounds on the norm. Sure, I understand there are standard reductions, but both these things need to be said somewhere. I mentioned this point as a "limitation", rather than as a "weakness", because your statement is very open to misinterpretation without any clarification, and has nothing to do with the quality of your own results.
> >
> > Regarding limitation 3: I'm not sure if it is okay to dismiss something as "obvious". If something is confusing to (or has the potential to confuse) the reader, then by definition, it is *not* obvious. It can never hurt to improve the presentation of the manuscript because the goal is not just to establish who did what first, but in the interest of science, it is also to spread and convey that knowledge successfully. Obviously, I'm not entitled to any improvement in your manuscript, but I'm just doing my job as a reviewer here. Please, see my comment about limitation 1 for this. In that whole example, I'm simply suggesting to give more context of the problem, as opposed to just stating numbers and expressions.
> >
> > I appreciate the authors responding to my comments/questions. I hope that they do make an attempt to understand my perspective here. Would be happy to clarify anything else.

---

> > > ### Author Response · Authors · 2022-08-08
> > > **Reply to Reviewer pHxC**
> > >
> > > Thank you for your helpful comments and clarifications! Now, we understand your questions better and will follow your suggestions.
> > >
> > > Question 1: We agree an explicit theorem will be better for showing our result for the worst-case scenario and will add one in Section 1.1.
> > >
> > > Limitation 1: We agree.  We should more explicitly define these terms in the paper by adding one line for each.
> > >
> > > Limitation 2: We will discuss the differences between the statistical setting (such as [KLSU19] and [a]) and empirical setting, as well as how statistical lower bounds imply empirical lower bounds.
> > >
> > > Limitation 3: We will point out that the trace-sensitive result can be obtained by setting $\tau=1$ in the tail-sensitive bound, thus the former is strictly no better than the latter. Also, we will rephrase the example to provide more context, such as adding more descriptions on the example's data distribution and how the clipping works on it.
> > >
> > > Any further comments are welcomed!
> > >
> > > [a] Kamath, G., Mouzakis, A., & Singhal, V. (2022). New Lower Bounds for Private Estimation and a Generalized Fingerprinting Lemma. arXiv preprint arXiv:2205.08532.

---

### Official Review · Reviewer_YXwa · 2022-07-14

**Rating:** 6
**Confidence:** 4
**Soundness:** 4 excellent
**Presentation:** 3 good
**Contribution:** 3 good

**Summary:**

The paper considers the problem of privately estimating the covariance matrix that corresponds to a dataset. Two algorithms are given, with each of them attaining better performance depending on the regime. The first algorithm is *trace sensitive*, in the sense that the error rate depends on the trace of the sample covariance matrix, which is equal to $\frac{1}{n} \sum_{i = 1}^n ||X_i||^2$. This algorithm yields a new worst-case upper bound on the error rate of $\frac{d^{\frac{1}{4}}}{\sqrt{n}}$ when $n^{\frac{2}{3}} \le d \le n^2$. The second algorithm is a *tail-sensitive* algorithm, in the sense that the error rate becomes worse when we have more data-points whose $\ell_2$-norm is close to $1$ (all points are assumed to exist in the unit ball).

**Questions:**

None for the time being.

**Limitations:**

The paper is a theoretical work and has no obvious societal impact.

**Strengths And Weaknesses:**

Strength: The problem considered in this work is central to private statistics. Out of the $2$ algorithms proposed in the paper, the one that I found of greater interest was the tail-sensitive one. I found the way SVT was employed in the context of the algorithm to be quite interesting.

Weaknesses: I did not identify any obvious flaws or weaknesses.

---

> ### Author Response · Authors · 2022-07-30
> **Reply to Reviewer YXwa**
>
> Thank you for your review and positive comments!

---

### Meta-Review · Area_Chair_2Rp3 · 2022-08-24

**Recommendation:** Accept
**Confidence:** Certain

**Metareview:**

The reviewers all concurred that the main result of this paper is quite interesting. It privately estimates the covariance better than established methods in particular parameter regimes. Given the clear accept sentiments towards this paper, there was little additional discussion.

**Award:**

No

---

### Decision · Program_Chairs · 2022-09-14

Accept